# Chitin Isolation and Chitosan Production from House Crickets (*Acheta domesticus*) by Environmentally Friendly Methods

**DOI:** 10.3390/molecules27155005

**Published:** 2022-08-06

**Authors:** Marios Psarianos, Shikha Ojha, Roland Schneider, Oliver K. Schlüter

**Affiliations:** 1Quality and Safety of Food and Feed, Leibniz Institute for Agricultural Engineering and Bioeconomy (ATB), Max-Eyth-Allee 100, 14469 Potsdam, Germany; 2Department of Bioengineering, Leibniz-Institute for Agricultural Engineering and Bioeconomy (ATB), 14469 Potsdam, Germany; 3Department of Agricultural and Food Sciences, University of Bologna, Piazza Goidanich 60, 47521 Cesena, Italy

**Keywords:** insect, chitosan, enzymes, fermentation, characterization, Resilient food system

## Abstract

Alternative methods were evaluated for chitin isolation from *Acheta domesticus.* Chemical demineralization was compared to fermentation with *Lactococcus lactis*, citric acid treatment, and microwave treatment, leading to a degree of demineralization of 91.1 ± 0.3, 97.3 ± 0.8, 70.5 ± 3.5, and 85.8 ± 1.3%, respectively. Fermentation with *Bacillus subtilis*, a deep eutectic solvent, and enzymatic digestion were tested for chitin isolation, generating materials with less than half the chitin content when compared to alkaline deproteinization. Chitosan was produced on a large scale by deacetylation of the chitinous material obtained from two selected processes: the chemical treatment and an alternative process combining *L. lactis* fermentation with bromelain deproteinization. The chemical and alternative processes resulted in similar chitosan content (81.9 and 88.0%), antioxidant activity (59 and 49%), and degree of deacetylation (66.6 and 62.9%), respectively. The chitosan products had comparable physical properties. Therefore, the alternative process is appropriate to replace the chemical process of chitin isolation for industrial applications.

## 1. Introduction

Edible insects are considered an important food source in many countries with approximately 2100 species consumed as food worldwide [1]. A change in food systems is essential to combat climate change [2], and edible insects are a strong candidate to replace conventional livestock due to their lower environmental impact [3] and good nutritional profile [4]. However, when taking advantage of alternative bioresources, all fractions of the edible insects must be used either for food or non-food applications.

One of the compounds in edible insects is chitin; the amount varies depending on the species, developmental stage, and age of the insect. For example, the chitin content from adult crickets and adult mealworms is estimated to be 67.1 and 137.2 mg/kg on a dry basis, respectively [5]. Chitin is a biopolymer that consists of β-(1-4)-N-acetyl-D-glucosamine [6] and is mainly found in invertebrates and fungal species [7]. Chitosan is a derivative of chitin that is formed through deacetylation (the conversion of chitin acetamide groups into amine groups) and has antioxidant [8], antitumor, and antimicrobial activities [9].

The common method for chitin isolation from insects consists of three major steps: delipidation, deproteinization, and demineralization. Usually, deproteinization and demineralization are performed with an alkaline and an acidic treatment, respectively, combined with heating [10]. This sequential chemical treatment is also applied to other materials, such as crustacean materials [11]. Nevertheless, this method has a major disadvantage of environmental pollution [12].

Several alternative methods have been suggested to replace the chemical treatments in crustacean materials, including fermentation [8], deep eutectic solvents (DES) [13], or less hazardous chemicals [14].

There is a growing interest regarding the house cricket (*Acheta domesticus*), owing to its relatively easy rearing process [15] and valuable nutritional profile [16]. However, there are limited studies aiming to extract chitin and evaluate chitosan from *A. domesticus*, with the exception of studies of the emulsifying properties of a chitin fraction [17] or the lipid binding and antimicrobial capacity of chitosan [18]. Consequently, the present study aims to test some of the alternative processes for chitin isolation from house crickets and to evaluate their applicability. Furthermore, an alternative process line for obtaining a chitinous material is designed and compared to a sequential chemical process, with respect to the properties of the chitosan generated after deacetylation of the chitinous product. Figure 1 presents the experimental setup used to compare and evaluate the different methods of chitin isolation in the present study.

## 2. Results and Discussion

### 2.1. Evaluation of Methods for Chitin Isolation

#### 2.1.1. Comparison of the %DDM

As shown in Figure 2, the use of different methods for demineralization of the insect flour had a significant effect (*p* > 0.05) on the efficiency of the process. The %DDM of each sample was significantly different (*p* < 0.05) from the one determined for the other samples. The chemical method resulted in 91.1 ± 0.3% demineralization, while the implementation of microwave treatment and fermentation led to 97.3 ± 0.8% and 85.80 ± 1.3% demineralization, respectively, while the use of citric acid led to only 70.5 ± 3.5% demineralization.

During microwave treatment, an oscillating electromagnetic field is initiated. Subsequently, a rotational displacement of polar molecules takes place, leading to the generation of heat and the transfer of ions and electrons [19]. The potential of microwave processing as well as fermentation for enhancing demineralization has also been demonstrated for crustacean materials such as prawn waste and shrimp shells [20,21].

#### 2.1.2. Evaluation of Chitin Content of Materials Subjected to Different Methods for Deproteinization and Chitin Extraction

Figure 3 presents the chitin content (mg/100 mg d.w.) of solid residues obtained after different deproteinization processes and one-step processes of for chitin isolation. The results showed no significant difference when using proteolytic enzymes, DES, or fermentation processes for chitin isolation. However, the material generated after treatment with NaOH had a significantly higher (*p* < 0.05) chitin content (27.1 ± 1.6 mg chitin/100 mg d.w. of the residue) than the other materials. The average chitin content of the other materials ranged between 6 and 13 mg/100 mg d.w. of the residue, with no significant differences (*p* > 0.05) among the chitin content of the different materials.

The implementation of the treatment of cricket flour with papain, bromelain, *Bacillus* species-generated proteases, and DES was not as successful, possibly because of differences in the nature of raw material compared to the crustacean materials; the chitin content of the material generated after each process was less than half of that of the material produced from the chemical treatment with NaOH.

This difference in efficiency of the methods based on other chitinous material can be attributed to the properties of the proteins present in the crickets, which have low extractability in aqueous solutions [22]. Furthermore, these proteins tend to form stable complexes with chitin, making complete deproteinization more difficult [23]. Da Silva, Brück & Brück (2017) [24] also reported residual impurities from fermented mealworm, despite a high degree of demineralization, which they did not identify as chitin, but rather as a chitinous material. However, enzymatic hydrolysis and DES have been successfully introduced to isolate chitin from *Tenebrio molitor* [24] and *Hermetia illucens* [25], respectively.

### 2.2. Selection of an Alternative Process for Chitin Isolation

Regarding the alternative method for chitin isolation, microwave-assisted demineralization was rejected to avoid the use of chemicals. The implementation of the DES, citric acid, and fermentation with *Bacillus subtilis* methods was considered inefficient. Finally, of the two enzymatic treatments, the one containing bromelain was considered more appropriate due to the shorter treatment time. The deproteinization step was performed before the demineralization, since it is reported that this sequence leads to a higher purity of chitin due to the removal of a protein layer that leaves chitin unprotected [26].

The chitin content of the isolated materials obtained through the chemical and the biological methods was 73.5 ± 0.2 and 56.3 ± 0.5 mg chitin/100 mg d.w. of isolated material, respectively. It is important to note that the chemical method, which is very commonly implemented for chitin isolation [10], did not lead to the isolation of pure chitin, as observed also by Kaya et al. (2015). Specifically, they argued that a degree of acetylation of chitin higher than 100%, which is reported for chitin obtained from other organisms, such as crab, shrimp, or bumblebee, is an indication of protein or mineral residues in the isolated chitin [27]. Regarding edible insects, there are only a few studies exploring the potential of an alternative and sustainable chitin isolation process. Fermentation has been applied to mealworms for the isolation of chitin with a high degree of demineralization (>90%) but also the presence of residual impurities [28].

### 2.3. Characterization of Chitinous Materials and Chitosan

#### 2.3.1. Properties of Chitosan

The properties of the produced chitosan materials are presented in Table 1. The chitosan content of Chitosans 1 and 2 appears to be high (81.9 and 88.9 mg chitin equivalent/100 mg d.w. of Chitosans 2 and 3, respectively), even though the chitin content of the solid materials that were isolated by both the chemical and biological extraction methods was lower. This is attributed to the extraction of protein residues during the deacetylation process, which would lead to the production of chitosan at higher purity than the chitin [29]. Further, the chitosan content of Chitosan 2 was significantly lower (*p* < 0.05) than that of Chitosan 3, which can be attributed to the partial dissolution during the acidic treatment [30]. Furthermore, the DD% of chitosan 1 is 75.1% and is higher than the chitosans produced from cricket flour (66.6 and 62.9% for chitosan 2 and 3, respectively), which both had a very similar value of DD%.

The chitosan that was produced from commercial chitin has four times greater (*p* < 0.05) molar mass compared to the chitin produced from crickets. Additionally, the chitosan samples obtained from the crickets have a low molecular weight, with that of Chitosan 3 being significantly lower (*p* < 0.05). The lower molar mass of the chitosan obtained from the biological process is attributed to the depolymerization of chitinous materials by bromelain. However, the low molar mass is a desirable property for chitosan, since lower molar mass indicates enhanced antioxidant, antimicrobial, and antitumor activities [31].

This indication was confirmed regarding the antioxidant activity. Specifically, Chitosan 1 had significantly lower (*p* < 0.05) antioxidant activity (%) than Chitosans 2 and 3. This result is attributed to the difference in molar mass between the samples. This result suggests a further advantage of implementing the biological method to generate the final chitosan product from the cricket flour.

#### 2.3.2. Characterization of Structure (FTIR)

The main bands identified from the spectra of the materials (Figure 4) were: N-H stretching at 3270 cm^−1^ (A), C-H stretching at 2880 cm^−1^ (B) of CH, CH_2_, and CH_3_ groups, C-O stretch of amide I at 1620 (C) and 1653 cm^−1^, N-H bend and N-C stretch at 1559 cm^−1^ (D), bending of CH_2_ and stretching vibration of C-N at 1420 and 1315 cm^−1^ (F), respectively, symmetrical deformation mode of CH_3_ at 1380 cm^−1^ (E), and anti-symmetric stretching the C-O-C bridge at 1156 cm^−1^ (G) [24,27,32].

Regarding the chitin isolated from the cricket flour by the chemical method, it was observed that due to the formed hydrogen bonds, the carbonyl groups (-C=O) and (-NH-) of amide I and II, respectively, appear at 1650 cm^−1^, and the one between the CH2OH and the carbonyl group (-C=O) appears at 1620 cm^−1^, meaning the band of amide I is split in two peaks. This indicates that the chitin in house crickets is α-chitin [32]. It is also observed that the O-H and N-H stretching bands that appear usually at 3450 and 3270 cm^−1^, respectively, overlap due to the water content and the peaks are not well separated [33]. Although the spectra of the chitosans are similar, the chitinous material that was isolated by the biological method appears different from the other chitinous fractions, with the peaks of amide I, N-H bend, and N-C stretch being higher due to the protein residues of the material.

#### 2.3.3. Thermal Stability (TGA)

For Chitosan 1, the TGA suggests that the decomposition takes place between 280 and 400 °C with a 45.2% mass loss (Figure 5a). Before the decomposition starts, the observed mass loss is 11.62%, which comprises a 6.5% mass loss between 0 and 68.8 °C and another 5.2% between 69 and 300 °C. For Chitosan 2 (Figure 5b), the decomposition takes place between 280 and 400 °C with a 46.3% mass loss. Before the decomposition starts, between 0 and 150 °C, there is a mass loss of 10.7%. For Chitosan 3 (Figure 5c), the decomposition takes place between 280 and 400 °C with a 49.6% mass loss. Before decomposition, the mass loss takes place between 0 and 150 °C and there is a 7.4% mass loss and then another 3.4% between 200 and 300 °C. The residual mass of all three samples was approximately 30%.

The output of the TGA of the three studied chitosan products is very similar to the one previously reported for chitosan. The mass change between 0 and 150 °C has been attributed to the evaporation of water and is usually less than 10%, while the decomposition has been reported to take place between 300 and 400 °C. However, a mass loss between 150 and 300 °C is less common and suggests a partial decomposition within that range of temperature as well. The residual mass has been reported to be approximately 30% at 600 °C [27,34,35]. The three samples had similar thermal stability.

#### 2.3.4. Morphology (SEM)

As seen in Figure 6, the two materials obtained from cricket flour have a smooth surface with pores of various diameters. The surface of the chitin obtained by the chemical method (Figure 6b) appears smoother than the material obtained by the biological method (Figure 6c). A similar morphology of chitin has been observed for *Zophobas morio* [36] and some body parts of *Argynnis Pandora* [37].

Chitosan 1 (Figure 7a) had a different morphology than the chitosan obtained from the isolated chitin from cricket flour (Figure 6b,c). Chitosan 1 had a flake-like structure. The chitosan produced from the cricket flour had a smooth surface, while some small pores appeared on the surface of Chitosan 3. The smooth surface of chitosan has been observed for chitosan from other species, such as cicada and grasshoppers [38], while a similar surface as the one depicted in Figure 7c has been observed for *Zophobas morio* [36].

## 3. Materials and Methods

### 3.1. Insect Flour and Defatting

A commercial cricket flour (Thailand Unique, Udon Thani, Thailand) was used for this study. The flour was initially defatted with n-hexane for 2 h. The amount of removed fat (17.7 ± 0.8 g fat/100 g d.w.) was determined gravimetrically, after removing the solvent with a rotary evaporator (Buchi R-100, Flawil, Switzerland). All chemicals were purchased from Carl Roth GmbH & Co. Kg (Karlsruhe, Germany), unless otherwise stated.

### 3.2. Chitin Isolation and Purification

#### 3.2.1. Deproteinization Processes

Chemical treatment and proteolytic enzymes were tested for deproteinization of defatted cricket flour.

##### Chemical Deproteinization

Briefly, the chemical deproteinization (DP1) was performed by mixing the flour with a 1 M NaOH solution at s/l ratio of 1:50 and agitating the mixture at 80 °C for 24 h [39,40].

##### Papain-Assisted Deproteinization

For the deproteinization with papain (DP2), the flour was mixed with a 5 mM cysteine solution (s/l = 1:20) and was digested through the enzymatic solution with an enzyme/substrate ratio of 1:100 (mg/mg). The pH was adjusted to 6.5 and the process was performed for 24 h at 60 °C [41]. The pH and temperature conditions were based on the specification provided by Carl Roth GmbH & Co. Kg (Karlsruhe, Germany) for commercial papain.

##### Bromelain-Assisted Deproteinization

The treatment with bromelain (DP3) was performed by mixing the flour with water (s/l = 1:20) and digesting it through the enzymatic treatment with an enzyme/substrate ratio of 2% w/w for 5 h at 60 °C after adjusting the pH value to 5.5 [42]. The pH and temperature conditions were based on the specification provided by Carl Roth GmbH & Co. Kg (Karlsruhe, Germany) for commercial bromelain.

#### 3.2.2. Demineralization Processes

Demineralization of defatted cricket flour was carried out using four different methods: chemical demineralization), microwave treatment, citric acid, and fermentation with Lactococcus lactis (DSMZ Braunschweig Germany, DSM 20729).

##### Chemical Demineralization

The chemical demineralization (DM1) refers to the mixing of the sample with 1 M HCl (s/l ratio 1:30) and then agitating for 2 h at 98 °C [27].

##### Microwave-Assisted Demineralization

For the microwave-assisted demineralization (DM2), the samples were added to a 1 M HCl solution (s/l ratio= 1:30) and then processed via microwave heating for 8 min at 500 W [20].

##### Citric Acid Demineralization

The demineralization with citric acid (DM3) was performed with 0.5 M citric acid (s/l ratio 1:30) and agitation for 2 h at room temperature [14].

##### Lactic Acid Demineralization

Fermentation with *L. lactis* spp. (previously stored in the form of lyophilized culture) was performed for demineralization (DM4). Bacterial strains of *L. lactis* were supplied by the library of the Department of Bioengineering and the Microbiology Lab of the Leibniz Institute for Agricultural Engineering and Bioeconomy, respectively. Sterile MRS Bouillon Lactobacillus broth acc. (Merck KGaA, Darmstadt, Germany) was mixed with the lyophilized culture that was cultivated at 30 °C for 48 h. The culture was composed of 10% *w*/*v* cricket flour. The amount of inoculant was <1% v/v. The fermentation lasted 7 days at 30 °C with agitation at 150 rpm [21].

#### 3.2.3. Single-Step Chitin Isolation

##### Fermentation with *B. subtilis*

Bacterial strain of *B. subtilis* was supplied by the library of the Department of Horticultural Engineering and the Microbiology Lab of the Leibniz Institute for Agricultural Engineering and Bioeconomy, respectively. The fermentation process with *B. subtilis* was performed in the following medium (g/L): peptone 10, yeast extract 5, and NaCl 5, with the pH adjusted to 7. The culture was composed of 5% (w/v) cricket flour and 5% (w/v) of glucose. The fermentation was performed for 5 days at 37 °C on a rotary shaker (150 rpm). The amount of inoculant was <1% *v*/*v* [8].

##### Eutectic Solvents

For chitin isolation by DES, choline chloride (Alfa Aesar, Massachusetts, United States) and malonic acid were used at a molar ratio of 1:1 with mixing for 2 h at 100 °C. Cricket flour was mixed with DES (3 g of flour with 50 g of CC/MA) and stirred at 80 °C for 3 h. Then, 100 mL of water was added and the mixture was stirred for another 30 min [13]. At the end of each process, the liquid phase was removed by centrifugation at 3200× *g*, 10 min, after being cooled to room temperature in a water bath. Then, the supernatant was removed and the pellet was washed with water, heated at 60 °C, and then centrifuged again. The washing was repeated until the pH of the water became neutral; then, the pellet was dried at 60 °C until it reached a constant weight, when it was retained for further analysis.

#### 3.2.4. Degree of Demineralization

The efficiency of each demineralization process was evaluated by estimating the ash content of the pellet and the initial material and calculating the degree of demineralization (DDM %) using Equation (1):(1)$$\%\mathrm{DDM}=\frac{\left[\left(M_{O\;}\cdot \;O\right)-\left(M_{R}\;\cdot R\right)\right]\cdot 100}{\left(M_{O\;}\cdot \;O\right)}$$
where *M_O_* and *M_R_* are the ash content (g/100 g d.w.) before and after demineralization, respectively, and *O* and *R* are the dry weight (g) of the material before and after the demineralization, respectively [8]. The ash content was determined gravimetrically by measuring weight before and after the samples were placed at 550 °C.

#### 3.2.5. Determination of Chitin Content

The efficiency of the deproteinization methods was evaluated by measuring chitin content of the isolated solid fraction [43]. Standard chitin was used for the calibration curve and the results were expressed as mg chitin/100 mg d.w.

### 3.3. Chitosan Production and Characterization

#### 3.3.1. Selected Processes for the Isolation of a Chitin-Rich Fraction

Two of the aforementioned processing pathways were selected for the production of a chitinous material. The first was the sequential chemical treatment of the defatted flour with NaOH (deproteinization) and HCl (demineralization), with washing steps in between; this was considered as the control. The second method was the sequential combination of the fermentation of the flour with *L. lactis* spp. and the proteolytic solution containing bromelain, with washing steps in between. Both processes were performed on a large scale with washing steps in between. The fermentation was performed in a 2L BIOSTAT B bioreactor (Sartorius AG, Germany) and continuous monitoring of the pH, which was initially equal to 5.9, dropped to 4.3 in the beginning of the 6th day, and remained stable until the end of the process. The obtained chitinous products were dried at 60 °C until constant weight. All treatments were performed as described in the previous section.

#### 3.3.2. Deacetylation of Products

Deacetylation of the two obtained chitinous materials, as well as of commercial chitin, was performed with a 50% NaOH solution for 3 h at 130 °C. The samples were cooled to room temperature and filtered. Then, the chitosan was washed with water until the washing water had a neutral pH [44]. Standard chitin was deacetylated similarly and chitosan was obtained from commercial chitin to be used as a reference for comparison.

#### 3.3.3. Characterization of Chitin-Rich Fractions

##### Chitosan Content

The chitin content of the generated materials, was determined spectrophotometrically, as described in Section 3.2.5 [43].

##### Fourier-Transform Infrared Spectrometry (FTIR)

The chitinous materials were analyzed directly by Fourier-transform infrared spectrometry (Nicolet iS5, Thermo Scientific, US-WI 53711 Madison, Waltham, MA, USA).

##### Scanning Electron Microscopy (SEM)

The morphology of the chitinous materials was analyzed by scanning electron microscopy (Phenom Elektronenmikroskop, Phenom-World BV, NL-5652 AM Eindhoven, The Netherlands).

#### 3.3.4. Characterization of the Produced Chitosan Materials

##### Chitosan Content

The chitosan content was determined as described for the chitin-rich fractions.

##### Fourier-Transform Infrared Spectrometry (FTIR)

The chitosan samples were analyzed as described for the chitin-rich fractions. The degree of deacetylation was calculated based on the FTIR signal from the following equations [45]:(2)DD%=100−DA%
A_1320_/A_1420_ = 0.3822 + 0.03133·DA(3)

##### Thermogravimetric Analysis (TGA)

The thermal stability of the obtained chitosan materials was determined by thermogravimetric analysis (TGA) with a STA449 F3 Jupiter instrument (NETZSCH-Gerätebau GmbH, D-95100 Selb, Germany) and a 10 °C/min temperature change from 25 to 650 °C in N_2_ [46].

##### Scanning Electron Microscopy (SEM)

The morphology of the obtained chitosan was analyzed as described for the chitinous materials.

##### Antioxidant Activity

The antioxidant activity of chitosan was determined with the DPPH radical scavenging assay. Specifically, 2,2-diphenyl-1-picrylhydrazyl (Alfa Aesar, Haverhill, MA, USA) was solubilized in methanol to obtain a DPPH solution of 6·10^−5^ M and the three chitosan products were solubilized in a 1% acetic acid solution to obtain chitosan solutions of 4.5 mg/mL. Then, 1 mL of chitosan solution was mixed with 3 mL of the DPPH solution and incubated at room temperature for 30 min in the dark. The absorbance was measured at 515 nm with a UV/Vis spectrometer (Spectronic Unicam UV1, Thermo Fisher Scientific, Waltham, MA, USA); 1 mL of the 1% acetic acid solution was used as a blank. Antioxidant activity was calculated from:Antioxidant activity% = (A_0_ − A)/A_0_(4)
where, A_0_ is the absorbance of the blank and A is the absorbance of the sample [27].

##### Determination of Molecular Weight

Viscometry was used to determine molecular weight. Chitosan solutions were prepared by stirring for 24 h at room temperature using a 0.1 M acetic acid/0.2 M NaCl solution with a concentration range of 0.2–1.5 g chitosan /L. The viscosity of the solutions was measured. Intrinsic viscosity [η] was calculated using Equation (5) and the molar mass was calculated using the Mark–Houwink equation (Equation (6)) [47]:
(5)$$[\eta ]={\left(\eta_{SP}/C\right)}_{C\to 0}={\left(\eta_{reduced}\right)}_{C\to 0}$$
(6)$$[\eta ]=K\;\cdot \;M_{w}^{a}$$
where K = 1.81 × 10^−3^ mL/g and α = 0.93 [48].

### 3.4. Statistical Analysis

Each experiment was repeated at least in triplicate. Significant differences among data obtained for samples generated by different processes were investigated with one-way analysis of variance (ANOVA). Tukey’s post hoc test was applied to separate means with significant differences (*p* ≤ 0.05). Data that did not follow a normal distribution were normalized before the analysis. The software used was IBM SPSS Statistics 23 (IBM Corp., Armonk, NY, USA).

## 4. Conclusions

The present study demonstrates that implementing alternative methods for chitin extraction from *Acheta domesticus* was efficient in removing minerals and less efficient in removing proteins compared to conventional treatments. Two chitinous materials were isolated using the conventional process and a biological process (digestion with bromelain solution and fermentation with *L. lactis*) and deacetylated into chitosan with comparable properties. Therefore, the biological process was appropriate for replacing the chemical process to generate a chitinous material as a basis for chitosan production from the cricket flour, with the advantage of non-hazardous waste. The successful large-scale production of chitosan following the biological method makes it applicable for industry. Regarding future perspectives, it is important to further investigate the properties of cricket-derived chitosan and explore its various properties and applications. In terms of the improvement of the extraction methods and the properties of the products, a variety of enzymes and fermentation techniques can be explored, as well as the effect of emerging technologies such as microwave treatment.

## Figures and Tables

**Figure 1 molecules-27-05005-f001:**
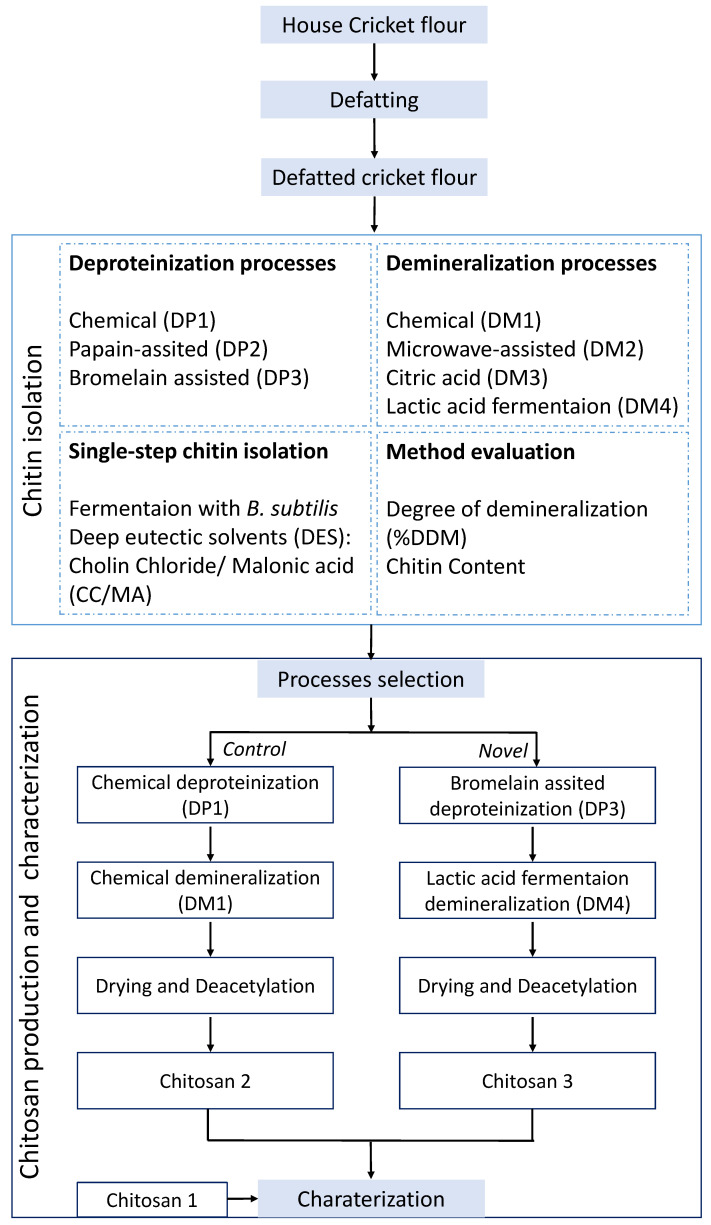
Flow chart of the experimental setup of the present study.

**Figure 2 molecules-27-05005-f002:**
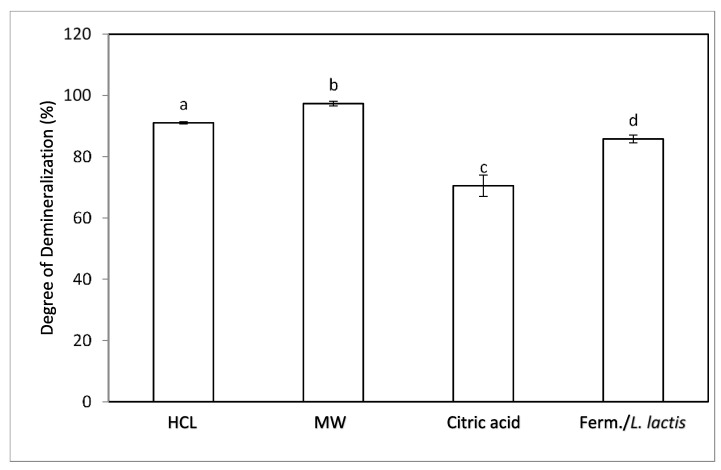
Efficiency of demineralization methods based on the degree of demineralization. Error bars indicate the standard error based on the variation in the %DDM. Different subscript letters indicate significant differences (*p* < 0.05) among the means of the %DDM obtained from each sample treated with a different demineralization process.

**Figure 3 molecules-27-05005-f003:**
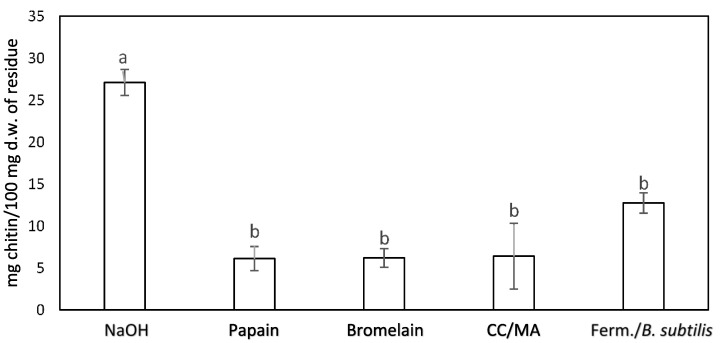
Chitin content (mg/100 mg d.w.) of solid residues obtained after different deproteinization processes or one-step processes for chitin isolation. Error bars indicate the standard error based on the variation in the chitin content. Different subscript letters indicate significant differences (*p* < 0.05) among the means of the chitin content determined for each sample.

**Figure 4 molecules-27-05005-f004:**
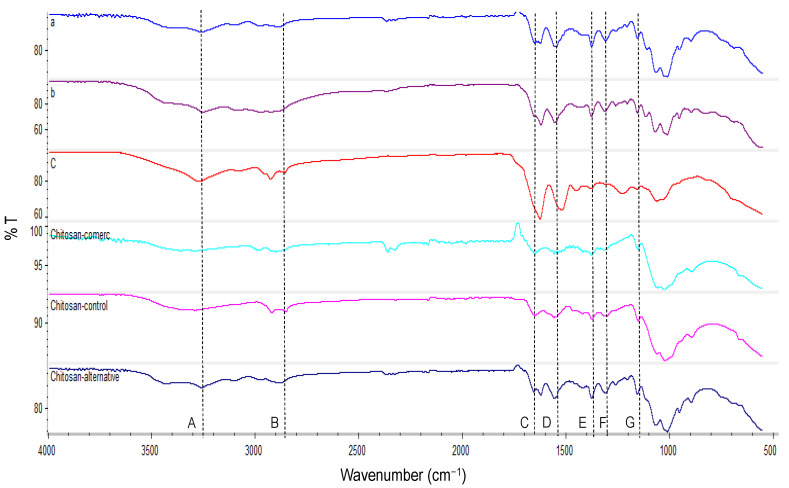
FTIR spectra of commercial chitin (a), chitin obtained from cricket flour by the chemical method (b), chitinous material obtained from cricket flour by the biological method (c), chitosan produced from commercial chitin (chitosan-commerc.), chitosan produced from chitin extracted from the cricket flour by the chemical method (Chitosan-control) and chitosan produced from chitin extracted from the cricket flour by the biological method (Chitosan-Alternative). A–G are the characteristic bands.

**Figure 5 molecules-27-05005-f005:**
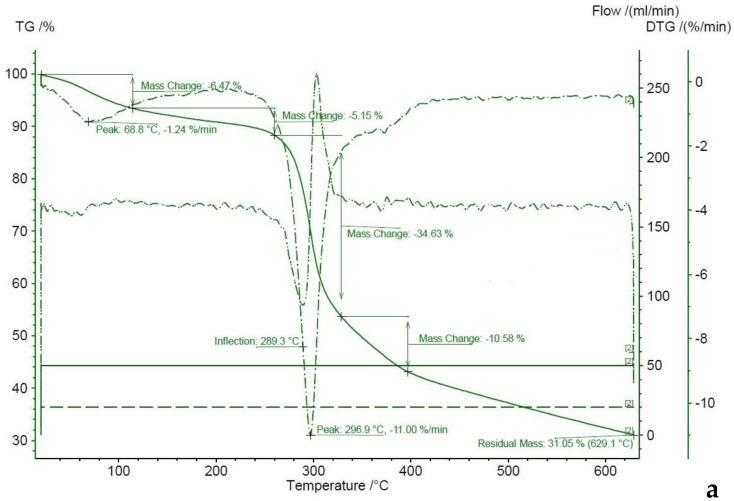
TGA thermographs and DTA curves of chitosan produced from commercial chitin (**a**), chitosan produced from chitin extracted from the cricket flour by the chemical method (**b**), and chitosan produced from chitin extracted from the cricket flour by the biological method (**c**).

**Figure 6 molecules-27-05005-f006:**
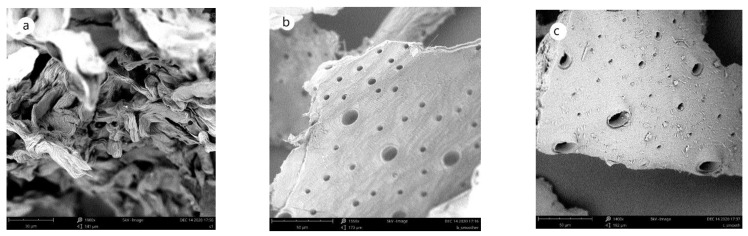
Morphology of chitin and chitinous material analyzed by SEM at 5 kV. (**a**) Commercial chitin, (**b**) chitin obtained from *Acheta domestica* by the chemical method, (**c**) chitinous material obtained from *Acheta domestica* by the biological method.

**Figure 7 molecules-27-05005-f007:**
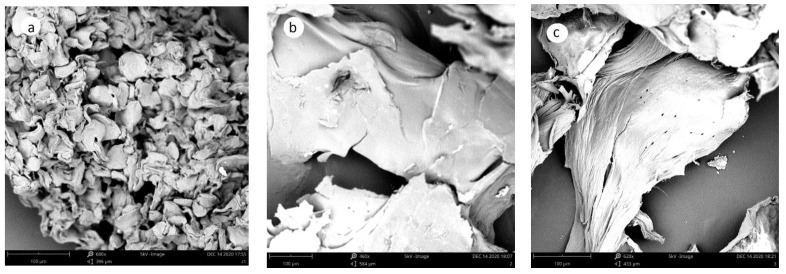
Morphology of chitosan analyzed by SEM at 5 kV. (**a**) Chitosan produced from commercial chitin, (**b**) chitosan produced from chitin obtained from *Acheta domestica* by the chemical method, (**c**) chitosan produced from chitinous material obtained from *Acheta domestica* by the biological method.

**Table 1 molecules-27-05005-t001:** Properties of obtained chitosan samples. Chitosan 1 refers to the chitosan produced from commercial chitin, Chitosan 2 to the chitosan produced from chitin isolated with the chemical method from cricket flour, and Chitosan 3 to the chitosan produced from the chitinous material isolated by the biological method from cricket flour. Data are presented as mean ± SD. Superscript letters (a, b, c) indicate significant differences (*p* < 0.05) between means of the same property of different chitosan samples.

Sample	Chitosan Content (mg Chitin Equivalent/100 mg)	Degree of Deacetylation (DD%)	Antioxidant Activity (%)	Molar Mass × 10^3^ (g/Mole)
Chitosan 1	-	75.1	35.00 ± 3.6 ^a^	471.2 ± 2.5 ^a^
Chitosan 2	81.9 ± 0.5 ^a^	66.6	59.0 ± 0.6 ^b^	103.4 ± 2.4 ^b^
Chitosan 3	88.0 ± 0.1 ^b^	62.9	49.3 ± 5.2 ^b^	86.8 ± 3.1 ^c^

## Data Availability

The data presented in this study are available on request from the corresponding author.

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
