# Peer review of "Chitin Isolation and Chitosan Production from House Crickets (Acheta domesticus) by Environmentally Friendly Methods"

_molecules, 2022, doi:10.3390/molecules27155005_

Round 1
Reviewer 1 Report
Following revisions need to be carried out:
1. Introduction section needs to be revised and elaborated further.
2. Results and discussion section should be strengthened in light with chemical explanation.
Author Response
Point by point response to reviewer's comments is attached

Reviewer 2 Report
Manuscript 1818932 describes the production of chitin from Acheta domesticus by providing a biological method. The manuscript needs to undergo revisions in accordance with the comments given below to increase its scientific impact.
1. Line 48, 382. Abbreviations should be used in the text for microorganism names, unless the complete name appears for the first time.
2. On page 7: The main and characteristic bands in the spectra should be labeled in figure 4.
3. On page 8: Some letters are not clear in figure 5, please provide the sharper figures.
4. The addition of XRD analysis is suggested to contrast the differences in crystallinity of chitin extracted by chemical and biological methods, since it is also a typical method to detect chitosan and chitin.
5. On page 9, in Figures 7 and 6, the magnification of the three images was inconsistent. To evaluate the morphological variations among several samples, please use SEM micrographs at the same magnification.
6. What about future work? Can you indicate one or two new strategies to improve chitosan products by Acheta domesticus?
Author Response

(The authors gave the same response as above.)

Reviewer 3 Report
Specific comments:
1. Throughout the text: The standard deviation should be expressed as ONE significant figure; that is, unless the number is between 11 and 19 times some power of ten, in which case you can use two significant figures. The mean value should be rounded off at the decimal place corresponding to the last significant digit of its standard deviation. E.g., ‘91.06±0.34, 97.33±0.79, 70.51±3.5 and 85.80±1.29%’ (lines 15-16) should be replaced with ‘91.1±0.3, 97.3±0.8, 71±4 and 85.8±1.3%’, etc.
2. Line 37: ‘substitution of the acetyl groups of chitin with amine groups’ gives -NH-NH2 groups, which is not the case. Rephrase it e.g. like this: ‘conversion of chitin acetamide groups into amine groups’.
3. Table 1: The molecular weight (in contrast to the molar mass) is unitless. Either remove g/mole or use term molar mass across the text.
4. Line 166: Replace ‘CH2 and CH2OH’ with ‘CH, CH2 and CH3’.
5. Line 171: Please rephrase this sentence. The 1650 and 1620 cm-1 absorption bands are not due to these hydrogen bonds, but the position of these bands depends on the hydrogen bonds formed.
Author Response

(The authors gave the same response as above.)

Round 2
Reviewer 1 Report
Accept
Reviewer 3 Report
The authors have successfully addressed all my concerns, improving the manuscript with their edits. In my opinion, the manuscript is now acceptable for publication.